# Validation of a Rhythmic Jumping Task for the Assessment of Upper–Lower Limb Coordination: Evidence from High-Level Athletes

**DOI:** 10.3390/jfmk10040473

**Published:** 2025-12-08

**Authors:** Runjie Li, Hitoshi Koda, Megumi Gonno, Toru Morihara, Tetsuya Miyazaki, Tomoyuki Matsui, Teruo Nomura, Kohei Okado, Chisato Yamamoto, Noriyuki Kida

**Affiliations:** 1Doctoral Programs of Biotechnology, Graduate School of Science and Technology, Kyoto Institute of Technology, Hashikami-cho, Matsugasaki, Sakyo-ku, Kyoto 606-8585, Japan; draculalee@icloud.com (R.L.); mtsports0512@gmail.com (T.M.); 2Faculty of Arts and Sciences, Kyoto Institute of Technology, Hashikami-cho, Matsugasaki, Sakyo-ku, Kyoto 606-8585, Japan; matsui.tomoyuki.sports.reha@gmail.com (T.M.); kida@kit.ac.jp (N.K.); 3Department of Childhood Education, Faculty of Childhood Education, Nagoya Aoi University, 3-40 Shioji-cho, Mizuho-ku, Nagoya-shi 467-8610, Japan; gonno@nagoya-aoi.ac.jp; 4Marutamachi Rehabilitation Clinic, 12 Nishinokyo Kurumazakacho Nakagyo-ku, Kyoto 604-8405, Japan; toru4271@koto.kpu-m.ac.jp; 5Department of Health and Sports Sciences, Faculty of Health and Medical Sciences, Kyoto University of Advanced Science, 1-1 Nanatani, Kameoka, Kyoto 621-8555, Japan; note0420@gmail.com; 6Faculty of Letters, Tohoku Gakuin University, 1-3-1 Tsuchitoi, Aoba-ku, Sendai 980-8511, Japan; okado@mail.tohoku-gakuin.ac.jp; 7Department of Rehabilitation, Rakuwakai Otowa Rehabilitation Hospital, 32-1 Koyama Kitamizo-cho, Yamashina-ku, Kyoto 607-8113, Japan; yama519chi@yahoo.co.jp

**Keywords:** rhythmic jumping task, upper–lower limb coordination, high coordination-level group, coordination ability, task generalizability, test–retest reliability

## Abstract

**Background:** A rhythmic jumping task that does not require specialized equipment may represent a simple method to assess upper–lower-limb coordination in athletes. Previous studies have been limited to groups with relatively low coordination ability; thus, whether task performance reflects the ability level or remains reproducible over time is unclear. This study determined whether the rhythmic jumping task reflects coordination levels in high-level performers, verified its generalizability as an assessment tool, and evaluated its reproducibility over time. **Methods:** Twenty-eight female high school volleyball players who routinely engaged in coordination training were enrolled, and performed six rhythmic jumping tasks identical to those used in a previous study. Performance was evaluated using three indices: complete performance rate (successful completion of all four series), success rate of at least one series, and average number of successful series. Twelve participants were retested 1 year later to examine reproducibility. **Results:** The high-level group demonstrated an overall superior performance compared to the low-level group from a previous study. Easier conditions yielded higher success rates, whereas more difficult conditions yielded lower success rates. Retest results demonstrated reproducible performance patterns over time. **Conclusions:** The rhythmic jumping task appropriately reflected coordination ability: high-level performers outperformed low-level performers. The task maintained a consistent difficulty order and reproducible performance across groups and over time, supporting its validity and generalizability as a practical and reliable tool to assess upper- and lower-limb coordination in applied athletic settings and provides a foundation for its further refinement and application as a standardized coordination assessment method.

## 1. Introduction

Motor coordination is the ability to perform intended movements by harmonizing multiple body segments and joints in both temporal and spatial domains [1,2,3]. It underlies a wide range of activities, from basic daily movements such as walking and stair climbing to complex skills such as playing the piano or performing athletic actions [4,5]. Developmental and motor-learning studies have demonstrated that individuals with higher motor coordination levels exhibit greater efficiency and precision in skill acquisition, leading to better outcomes in motor learning and athletic performance [6].

Various methods have been developed to assess motor coordination quantitatively. For instance, precise measurement techniques, such as three-dimensional motion analysis systems and electromyography, have enabled detailed analysis of movement patterns [7]. Conversely, several field-based evaluation tasks that require no specialized equipment have been proposed. Tasks such as hopscotch, single-leg standing, and mini-hurdle running have been reported to effectively assess coordination ability in developing children [8,9,10]. Nevertheless, these approaches are primarily designed for children in developmental stages and are not suitable for adults or athletes. Consequently, simple and practical assessment methods that can directly evaluate motor performance in adult and athletic populations are lacking.

A simple field-based test using a rhythmic jumping task was proposed in the previous study to evaluate upper–lower limb coordination [11]. Specifically, participants were instructed to maintain a constant arm movement while manipulating the direction and rhythm of their lower-limb movements. In that study, which targeted male high-school baseball players, three major factors that made upper–lower limb coordination difficult were identified: (1) when the movement directions of the upper and lower limbs were incongruent; (2) when one lower-limb movement per beat was combined with two upper-limb movements; and (3) when one lower-limb movement per beat was performed with no arm movement. The proposed task allows the difficulty level to be adjusted stepwise by combining these elements. Because it does not require special equipment and can be easily implemented in practical field settings, its usefulness as a coordination assessment tool has been demonstrated.

However, that study targeted baseball players who had not received specialized upper–lower limb coordination training [11]. Because athletes without specific coordination training typically exhibit lower proficiency in the corresponding motor abilities [6,12], the study participants could reasonably be regarded as a lower-level group for upper–lower limb coordination. Accordingly, an important unresolved issue is whether the rhythmic jumping task maintains its difficulty structure and discriminatory characteristics when applied to groups with substantially higher coordination proficiency. In other words, it remains unclear whether the observed difficulty pattern is specific to that particular population or generalizable across different coordination levels. For instance, whether the three major sources of upper–lower limb coordination difficulty identified in the previous study similarly apply to individuals with higher coordination ability has not yet been clarified. In addition, whether individuals with higher coordination ability indeed achieve superior performance also remains uncertain, which is important for determining the discriminant validity of the task. Furthermore, outcome stability is critical for any assessment, as large fluctuations upon retesting would undermine its utility. Although a certain degree of stability was reported in the previous study [11], whether similar stability can be confirmed across different populations remains unknown. Addressing these uncertainties is essential for a deeper understanding of the utility of the assessment methods.

Therefore, the primary aim of this study was to clarify whether the rhythmic jumping task developed in the previous study possesses sufficient validity as a practical assessment method to assess upper–lower limb coordination in applied settings [11]. Three specific objectives were established. First, this study aimed to examine construct and discriminant validity by determining whether the previously proposed test could appropriately reflect differences in coordination ability when administered to athletes assumed to have a high level of upper–lower limb coordination. Second, this study sought to verify whether the three factors identified in the previous study as making upper–lower limb coordination difficult remained consistent among individuals with higher coordination ability. Third, this study aimed to evaluate the stability of test performance over an approximately one-year interval, in order to explore whether the RJT reflects relatively stable, trait-like aspects of coordination rather than short-term fluctuations.

Taken together, establishing whether the RJT maintains its validity and stability across different coordination proficiency levels is essential for determining its broader applicability. Because the task requires no specialized equipment and can be implemented easily in field settings, confirming its robustness would support its potential use not only in school sports and community or competitive teams but also in rehabilitation contexts, particularly in environments with limited access to laboratory-grade measurement tools.

## 2. Materials and Methods

### 2.1. Participants

Twenty-eight female high school volleyball players participated in this study (age: 15.9 ± 0.8 years; height: 163.5 ± 6.0 cm; body mass: 58.6 ± 5.0 kg). All participants routinely engaged in sports-specific training as part of their competitive activities, including continuous exercises that required upper–lower limb coordination. None of the participants had a history of neurological or musculoskeletal disorders. Among them, 12 players participated in a follow-up measurement approximately 1 year after the initial testing. The remaining 16 players were unable to participate in the follow-up sessions because they left the team after graduating from high school.

In the previous study, the participants were male high school baseball players [11]. To maintain consistency in participant age, the present study was directed at high school athletes. Because volleyball involves coordinated upper–lower limb movements similar to those required in baseball, female volleyball players were selected as participants. Previous studies have shown that rhythmic movement training enhances interlimb synchronization and stabilizes coordinated performance through sensorimotor entrainment mechanisms [13,14,15]. Therefore, volleyball players who routinely engage in rhythm-integrated drills as part of their sports-specific practice were considered to represent the high-coordination-level group in this study. The number of participants was determined by the number of players available during the data-collection period. To ensure sampling consistency and minimize variability in athletic experience, all participants were recruited from the same team.

### 2.2. Task Structure, Motion Definitions, and Implementation Procedure

The task used in this study was the rhythmic jumping task (RJT), which was developed and validated in the previous study [11]. This task requires coordinated movement of the upper and lower limbs. All motions were restricted to the frontal plane and performed using laterally symmetrical movement patterns, allowing the RJT to evaluate upper–lower limb coordination under standardized, simplified conditions. The participants performed light, continuous jumps in synchrony with a metronome beat, coordinating both upper- and lower-limb movements. Six tasks identical to those employed in the previous study were administered. The metronome tempo was set at 120 beats per minute (bpm), as in the previous study. This tempo range (100–120 bpm) has been reported to represent a natural and stable rhythm for performing rhythmic movements in both adults and high-school students [11,16]. The posture assumed at the moment of each metronome beat was defined as “posture,” and the arm and leg movements executed between beats were described as “motion.” Four motions constituted one series, and four consecutive series comprised one task. To address the reviewers’ request for a complete description of all six tasks, Figure 1 illustrate the representative motion sequences for Tasks 1 to 6. This illustration were created based on the movement definitions established in the previous study [11]. As shown in the figure, the upper-limb movement pattern was identical across all tasks. Participants began with their arms resting naturally at their sides. In synchrony with the metronome beat, both hands moved upward toward the center above the head and reached the top position exactly on the beat. The hands then moved downward toward the shoulders, making contact with the ipsilateral shoulder on the second beat. The hands remained on the same-side shoulders until the third beat. Finally, both hands moved rapidly to the contralateral shoulders and returned to the ipsilateral shoulders on the fourth beat, completing one series of upper-limb movements. Without returning to the ready posture, the hands immediately moved upward again to initiate the next series.

The lower-limb movements differed across the six tasks while sharing the same ready posture: feet together. Each task consisted of four motions forming one series, and participants performed four consecutive series to complete one task. For instance, in Task 1 (Figure 1), one series consisted of the following sequence: a jump while keeping the feet together, a jump with the feet apart, and two additional jumps maintaining the feet-apart posture. A new series began with a feet-together jump. Tasks 2 through 6 followed the same structural rule but incorporated different predefined lower-limb combinations, which are illustrated in Figure 1.

To ensure comparability with the previously analyzed group of lower-coordination participants, the same procedural steps were followed. A 5-min practice period, identical to that used in the previous study, was provided to balance practice and rest, thereby promoting efficient motor learning, preventing excessive fatigue, and maintaining an overall experimental schedule [11,17].

### 2.3. Data Processing

All data were obtained from video recordings captured during the experimental sessions using a digital video camera (GC-YJ40; JVC Kenwood Corporation, Yokohama, Japan, manufactured in Malaysia). The recordings were reviewed on a personal computer (MacBook Pro; Apple Inc., Cupertino, CA, USA) with the default QuickTime Player for macOS, and all statistical analyses were conducted using IBM SPSS Statistics (version 29.0; IBM Corporation, Armonk, NY, USA).

To ensure accuracy and reduce potential evaluator bias, two researchers were involved in the scoring process. One researcher initially entered the data by reviewing each recorded trial and judging the success or failure of each series, and a second researcher independently re-checked all entries. Any inconsistencies were resolved through joint re-examination of the corresponding video segments. Success and failure were judged according to a checklist developed based on the predefined movement criteria described in the previous study [11]. Each series served as the minimum analytical unit, and a series was considered successful when the performed motion clearly corresponded to the prescribed movement pattern. The checklist confirmed key items such as whether the four upper-limb movements were executed in the correct order and direction, whether the prescribed apart–together pattern of the lower limbs was maintained, whether the timing of takeoff and landing aligned with the metronome beats without premature or delayed responses, whether the hand and foot positions at each beat matched the task definitions, and whether unintended additional motions occurred during the jumps. Because the rhythmic jumping task was designed as a simple field-based assessment that can be evaluated visually, the criteria allowed a limited range of variation in movements when the intended pattern was clearly maintained.

Three evaluation indices defined in the previous study [11] were adopted here: the success rate in completing all four series, the success rate in completing at least one series, and the average number of successful series. Each series served as the minimum analytical unit, and success or failure was judged according to previously established criteria; that is, a series was regarded as successful when the performed movement pattern corresponded precisely to the prescribed task motion. Detailed definitions of these evaluation indices and the success–failure criteria can be found in the previous publication [11]. Statistical comparisons among the six tasks were examined using Cochran’s Q test and paired *t*-tests, whereas longitudinal changes across approximately one year were assessed using McNemar’s test, Cramér’s V, and Spearman’s rank correlation coefficient (ρ). A significance threshold of 5% was used for all the analyses.

## 3. Results

### 3.1. Comparison of the Task Performance Across the Six Tasks

Comparisons across the six tasks are shown in Figure 2. Figure 2A displays the proportion of participants who completed all four series, along with the results of comparisons between tasks. The success rates for Tasks 1 (82.1%) and 2 (82.1%) were significantly higher than those for Tasks 4 (28.6%), 5 (25.0%), and 6 (14.3%) (all *p* < 0.05). Although the success rate for Task 3 (46.4%) was lower than that for Tasks 1 and 2, the difference was not statistically significant (*p* = 0.053). No significant differences were found between Tasks 3, 4, 5, and 6.

Figure 2B presents the proportion of participants who completed at least one series, together with the results of the comparisons between tasks. The success rates for Tasks 1 (100.0%) and 2 (100.0%) were the highest among all tasks and were significantly higher than those for Task 6 (71.4%) (*p* = 0.006 for both). No significant differences were found between Tasks 1 and 5 or between Tasks 3 and 6.

Figure 2C shows the mean number of successful series for each task, together with the results of comparisons between tasks. The mean number of successful series for Task 1 (3.75 ± 0.65 series) and Task 2 (3.68 ± 0.82 series) was the highest among all tasks, with no statistically significant difference between them (*p* = 0.602). In contrast, the mean number of successful series for Tasks 1 and 2 was significantly higher than that for Tasks 3 (*p* < 0.001 and *p* = 0.002, respectively), 4 (both *p* < 0.001), 5 (both *p* < 0.001), and 6 (both *p* < 0.001). The mean number of successful series for Task 6 (1.39 ± 1.37 series) was the lowest among all tasks and was significantly lower than those for Tasks 1–5 (all *p* < 0.05). Additionally, the mean number of successful series for Task 3 was significantly higher than that for Tasks 5 (*p* = 0.039) and 6 (*p* < 0.001), whereas no significant difference was found between Tasks 3 and 4 (*p* = 0.055) or between Tasks 4 and 5 (*p* = 0.568).

### 3.2. Comparison of First and Second Session Participants

Figure 3 summarizes the performance comparison between the first and second sessions for the same participant across six tasks. Figure 3A illustrates the success rates based on the criterion of completing all four series, which were compared between the first and second measurements using McNemar’s test. For Task 1, the success rate decreased from 91.7% in the first measurement to 75.0% in the second, with no significant difference (χ^2^ = 0.000, *p* = 1.000). Task 2 demonstrated no success rate changes between the first and second measurements (83.3% → 83.3%; χ^2^ = 0.000, *p* = 1.000). For Task 3, the success rate increased from 41.7% to 58.3%; nonetheless, the difference was not significant (χ^2^ = 1.371, *p* = 1.000). In Task 4, the success rate increased from 25.0% to 50.0%, with no significant difference (χ^2^ = 0.043, *p* = 1.000). Task 5 and Task 6 exhibited an increase from 16.7% to 41.7% (χ^2^ = 1.097, *p* = 1.000) and from 14.3% to 50.0% (χ^2^ = 0.000, *p* = 1.000), respectively, neither of which demonstrated a significant difference. No significant differences were observed in any of the tasks completed between the first and second measurements.

Figure 3B presents the success rates based on the criterion of completing at least one series, which was compared between the first and second measurements using McNemar’s test. For Tasks 1 and 2, all participants succeeded in both the first and second measurements (100.0% → 100.0%); therefore, statistical testing could not be performed. For Tasks 3, 4, 5, and 6 the success rate increased from 91.7% to 100.0% (χ^2^ = 0.000, *p* = 1.000), 83.3% to 100.0% (χ^2^ = 0.500, *p* = 1.000), 83.3% to 100.0% (χ^2^ = 0.500, *p* = 1.000), and 75.0% to 83.3% (χ^2^ = 0.000, *p* = 1.000), respectively. None of these changes was statistically significant.

Figure 3C displays the mean number of successful series for each task, which were compared between the first and second measurements using paired t-tests. Significant increases were observed for Task 4 (2.50 ± 1.62 → 3.58 ± 0.67; t = −2.238, *p* = 0.047), Task 5 (1.58 ± 1.31 → 2.75 ± 1.22; t = −3.189, *p* = 0.009), and Task 6 (1.50 ± 1.45 → 2.83 ± 1.59; t = −2.966, *p* = 0.013). No significant differences were found for Tasks 1 to 3 (t = 1.149, *p* = 0.275; t = −0.616, *p* = 0.551; and t = −1.254, *p* = 0.236, respectively).

The associations between the first and second measurements were examined. Based on Cramér’s V for success across all four series, a significant association was found only for task 6 (V = 0.707, *p* = 0.014). No significant associations were observed for Tasks 1 (V = 0.289, *p* = 0.317), 2 (V = 0.000, *p* = 1.000), 3 (V = 0.167, *p* = 0.564), 4 (V = 0.471, *p* = 0.102), or 5 (V = 0.500, *p* = 0.083). Next, based on the results of Cramér’s V for success in at least one series, no significant associations were found for Tasks 3 (V = 0.289, *p* = 1.000), 4 (V = 0.289, *p* = 0.615), 5 (V = 0.289, *p* = 0.500), or 6 (V = 0.408, *p* = 0.086). Statistical tests could not be performed for Tasks 1 and 2 because all participants succeeded in both the first and second measurements. Finally, the results of Spearman’s rank correlation coefficients for the average number of successful series showed positive correlations for Tasks 3 (ρ = 0.550, *p* = 0.064) and 6 (ρ = 0.547, *p* = 0.066), although these did not reach statistical significance. No significant correlations were found for the other tasks.

## 4. Discussion

### 4.1. Task Performance in the High-Level Upper and Lower Limb Coordination Group

Task success rates were analyzed in participants classified as having a high level of upper–lower limb coordination ability. For the indicator representing success across all four series, Tasks 1 and 2 demonstrated achievement rates exceeding 80%, Task 3 approached 50%, Tasks 4 and 5 were approximately 30%, while Task 6 slightly exceeded 10%. For the indicator of success in at least one series, all participants succeeded in Tasks 1 and 2 and maintained an achievement rate of over 60% in the remaining tasks. Regarding the average number of successful series, more than two successful series were achieved in all tasks, except Tasks 5 and 6. Conversely, the previous investigation reported that participants with a lower level of coordination ability achieved success rates below 50% even for Task 1, approximately 20% for Tasks 3 and 4, and nearly zero for Tasks 5 and 6. For the “success in at least one series” indicator, only a small proportion of participants succeeded in a single series for Tasks 1 and 2, and all other tasks were below 60%. Furthermore, the same study indicated that, in terms of the average number of successful series, only Tasks 1 and 2 exceeded two successful series, whereas several participants failed to achieve even one successful series in Tasks 4–6. Overall, the present results demonstrated higher success rates across all indices compared to those reported previously [11]. Differences in performance levels are primarily attributable to higher coordination ability among the current participants; nonetheless, possible influences of factors such as sports-specific characteristics should also be considered. Regarding sports-specific characteristics, although the groups differed in athletic background, volleyball and baseball share fundamental components, such as perceptual–motor coupling and interpersonal coordination within team contexts, suggesting that task demands are comparable [18,19]. Furthermore, anthropometric factors, including body mass index and physique, exhibit only a weak relationship with coordination task performance [20,21]; therefore, potential differences in body size between the two groups are unlikely to have been a major source of variation. Instead, the primary determinant of the observed differences appears to be whether participants engaged in regular, long-term upper limb coordination training in their daily routines. Previous literature has demonstrated that systematic and deliberate practice enhances coordination ability and produces transfer effects on related motor and cognitive tasks [22,23,24]. Thus, the participants can be regarded as a high-level coordination group, and their superior performance accurately reflects this status. Moreover, no “reversal phenomenon,” in which individuals with low coordination ability outperform those with higher ability, was observed, supporting the validity of the task as an appropriate measure for distinguishing levels of coordination ability. However, because volleyball players regularly engage in rhythm-based and whole-body coordination training as part of their daily practice, their performance in the present study may have been positively influenced by sport-specific training characteristics. This possibility should be considered when interpreting differences relative to the baseball players examined previously.

The task difficulty order was analyzed based on the success rates of the six coordination tasks. Regarding the indicator of success across all four series, Tasks 1 and 2 demonstrated higher success rates than did Tasks 3–6. Among Tasks 3–6, a gradual decline in success rate was observed, with Task 6 demonstrating the lowest value. Similarly, for the average number of successful series, Tasks 1 and 2 demonstrated a greater number of successful series than did the other tasks, and a stepwise decrease in the number of successful series was observed across Tasks 3–6. In contrast, for the indicator of success in at least one series, Tasks 1 and 2 had the highest success rates, while Task 6 exhibited the lowest. Nonetheless, among Tasks 3–5, the trend was not a simple stepwise decrease, as Task 5 exhibited a relatively higher success rate. When these three indicators were considered together, Tasks 1 and 2 were identified as the easiest to accomplish; Tasks 3 and 4 were moderately difficult; and Task 5 was relatively easy to perform as a single series but showed a lower success rate when performed continuously. Task 6 was challenging in both single- and consecutive-series conditions and was thus identified as the most difficult task. Notably, although there were differences in the absolute success rates, the overall order of task difficulty was consistent with that reported in the previous investigation [11]. Specifically, the pattern whereby Task 5 exhibited a relatively higher achievement rate on the indicator of at least one successful series among Tasks 3–5 was also consistent with the findings of the earlier report. The previous study identified three major factors that contribute to task difficulty. The first common difficulty observed across all tasks was the disparity in movement frequency between the upper and lower limbs; the upper limbs performed two movements per beat, whereas the lower limbs performed one. This often caused the lower limbs to synchronize by performing two movements when the upper limbs did two, or, conversely, caused the upper limbs to pause after one movement to match the rhythm of the lower limbs. The second difficulty occurred when the upper limbs were required to remain still during lower-limb movement, as involuntary upper-limb motion could result from transfer of the lower-limb movement. The third difficulty involved directional incongruence between the upper- and lower-limb movements, which led to failure of coordinated control. Specifically, when consecutive series were performed, mismatched movement directions between the series tended to increase the likelihood of failure. Considering these difficulty factors, among the six tasks, Task 1 contained none of the additional difficulty elements beyond the common ones and was therefore the simplest. Contrastingly, all movement patterns in Task 6 incorporated these difficulty factors, resulting in the lowest success rate. Regarding Task 5, when a single series was performed, the absence of the difficulty factor—specifically, the condition of keeping the upper limbs static during a one-beat lower-limb motion—made the task easier than Tasks 3, 4, and 6. Nevertheless, during consecutive performances, Task 5 shared the same characteristics as Task 6, wherein mismatched directions of the upper- and lower-limb movements between series reduced the success rate. Therefore, although participants’ attributes and coordination ability levels differed between the present and previous investigations, the difficulty structure derived from the task design remained consistent. This observation aligns with the notion that coordination task complexity depends primarily on task structure rather than on individual characteristics [25]. Accordingly, the present task appears to exhibit a consistent difficulty structure in this population. Nevertheless, the similarity in difficulty patterns between the two studies may, at least in part, reflect structural characteristics inherent to the RJT itself rather than broad validity across distinct populations. Further validation across additional sports, age groups, and cultural contexts will be necessary before strong claims regarding generalizability can be made.

Furthermore, among participants who underwent retesting 1 year after the initial measurement, overall performance was either comparable to or slightly improved relative to the previous results. No substantial decline or reversal in the order of task achievement was observed across tasks. The previous longitudinal investigation [11] similarly reported that after 1 year, no significant deterioration or reversal of the performance order occurred. These findings suggest that the task produced minimal temporal fluctuations in performance and maintained a high level of consistency in assessment over a certain period. Accordingly, this task shows promise as a tool for monitoring longitudinal changes; however, given the limited retest sample size, these findings should be interpreted with caution. In addition, although most retest differences were not statistically significant, subtle learning effects between the two sessions cannot be ruled out, particularly given the small retest sample size.

Three principal findings were obtained in the present analysis. First, participants with higher upper–lower limb coordination ability achieved higher task scores, and did not show paradoxical “reversal phenomenon” in which those with lower ability achieved higher scores was observed. Second, the order of task difficulty remained consistent across the different participant groups and corresponded with the trend reported in the previous investigation [11], suggesting that the observed difficulty patterns may reflect structural characteristics inherent to the task design. Third, in the retest conducted 1 year later, overall performance levels were maintained or improved, with no reversal in task order or notable decline, suggesting relatively stable performance patterns over time. Collectively, these findings support the potential usefulness of the present task as an assessment tool that may reflect differences in upper–lower limb coordination ability and may be suitable for continuous evaluation over time.

### 4.2. Limitations

This study had some limitations. First, both the present study and previous studies targeted athletes engaged in ball games, which share common elements such as perceptual–motor coupling and interpersonal coordination within team contexts. Therefore, it is necessary to examine the extent to which the validity of this task can be maintained across populations with substantially different coordination characteristics, including sports that require extremely high coordination, such as rhythmic gymnastics, and those with relatively low coordination demands, such as weightlifting. Moreover, because the present study did not include participants from different cultural or age groups, the generalizability of the task structure across broader populations remains to be verified. Second, task performance was evaluated through observational analysis of video recordings without the use of physiological measures, such as electromyography or three-dimensional motion analysis. Therefore, this study did not directly examine the neurological mechanisms underlying coordination control. In addition, although two evaluators independently checked all recordings, the use of video-based judgement may still introduce a degree of subjective bias compared with instrumented biomechanical analyses. Thus, criterion validity was not examined in this study.

Third, the sample size was limited, and further investigations involving larger and more diverse participant groups are warranted to generalize the findings. The sample size became even smaller in the retest session owing to attrition (12 of 28 participants), which further reduced statistical power and made it difficult to detect subtle longitudinal changes. Furthermore, the study did not employ randomization or include a control group, and thus causal inferences regarding training effects or group differences cannot be drawn. In addition, because multiple statistical comparisons were conducted across the six tasks and several performance indicators, the possibility of inflated Type I error cannot be fully ruled out, even though the analyses were hypothesis-driven. Therefore, the interpretation of *p*-values should be made with caution.

Additionally, factors such as the testing environment, individual athletic history, and lifestyle habits, which were not fully controlled in this study, may have influenced the results. Therefore, it is necessary to further clarify the evaluation characteristics and applicability of this task through multifaceted examinations that consider its potential limitations.

### 4.3. Future Directions

This study was significant because it targeted a group with high upper–lower limb coordination abilities, which had not been examined in the previous study [11]; moreover, it comprehensively investigated the universal evaluation characteristics of the task from multiple perspectives. The consistent order of task difficulty observed across groups with different ability levels and sporting backgrounds demonstrates that the task appropriately reflects coordination ability. This finding complements the results of previous studies and substantiates the general applicability and validity of this task as an evaluation method. Therefore, we propose several directions for future research.

First, the target population should be expanded to include athletes who participate in sports with distinctly different coordination demands as well as general adolescents, adults, and older adults to verify the applicability and validity of the task across a wider range of groups.

Second, by combining physiological indicators, including electromyography and three-dimensional motion analysis, future research may elucidate the relationships between task performance, motor control, and neural mechanisms, generating a more detailed understanding of the coordination strategies that support task achievement.

Third, by focusing on the “potential for creation” inherent in this task and identifying movement factors that determine difficulty structure, it will be possible to design new tasks with precisely adjusted difficulty levels tailored to the challenge levels of different populations. Additionally, this approach is expected to be effective in identifying differences within high-ability groups that may not be detected using existing tasks.

Through these efforts, it is expected that the findings from this study regarding the universality and stability of the task will be further advanced, contributing to the establishment of a comprehensive evaluation system for the assessment and long-term monitoring of coordination between the upper and lower limbs.

## 5. Conclusions

The present study was conducted as an extension of a previously proposed evaluation task to assess upper–lower limb coordination, targeting individuals with higher levels of coordination ability. This research aimed to further examine the validity and appropriateness of the proposed task as an assessment method. A comparison with the previous study involving participants with lower coordination levels revealed that individuals with higher coordination ability achieved consistently higher success rates across all tasks without any reversal phenomena. Thus, the evaluation task appropriately reflects the level of upper–lower limb coordination ability.

Furthermore, the order of task difficulty remained consistent across groups with different ability levels, demonstrating that the difficulty structure is independent of individual performance attributes and thus reflects a universal characteristic of the task. Moreover, the follow-up assessment conducted after 1 year demonstrated no significant decline or reversal in task performance, suggesting that the evaluation exhibited temporal stability and reproducibility.

Overall, these findings demonstrate that the proposed task possesses favorable characteristics in differentiating coordination ability and maintaining evaluative consistency, further supporting its potential as an assessment tool for upper–lower limb coordination. Although certain limitations remain, such as the restricted sample size and absence of physiological indices, the present study provides an important foundation for further verification in more diverse populations. Overall, by confirming the validity, universality, and stability of the task, this study offers empirical evidence for the establishment of a sustainable evaluation framework for coordination abilities.

## Figures and Tables

**Figure 1 jfmk-10-00473-f001:**
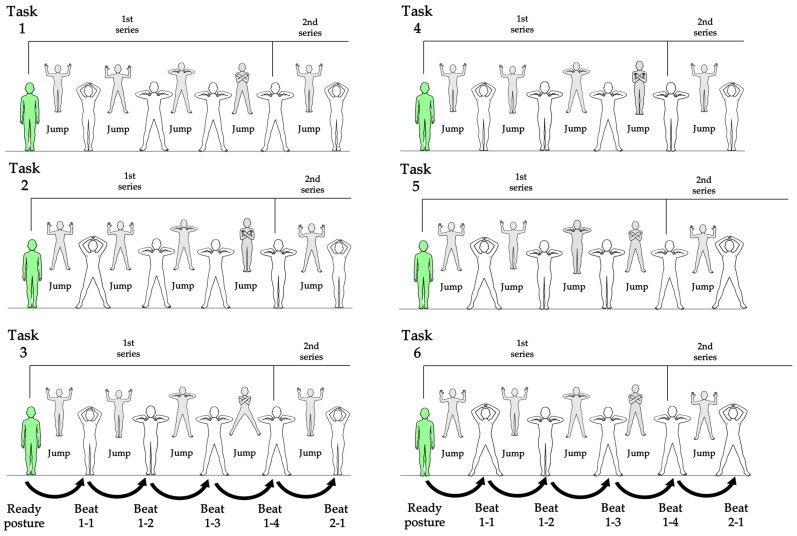
Images of upper and lower limb motion patterns in Task 1–6.

**Figure 2 jfmk-10-00473-f002:**
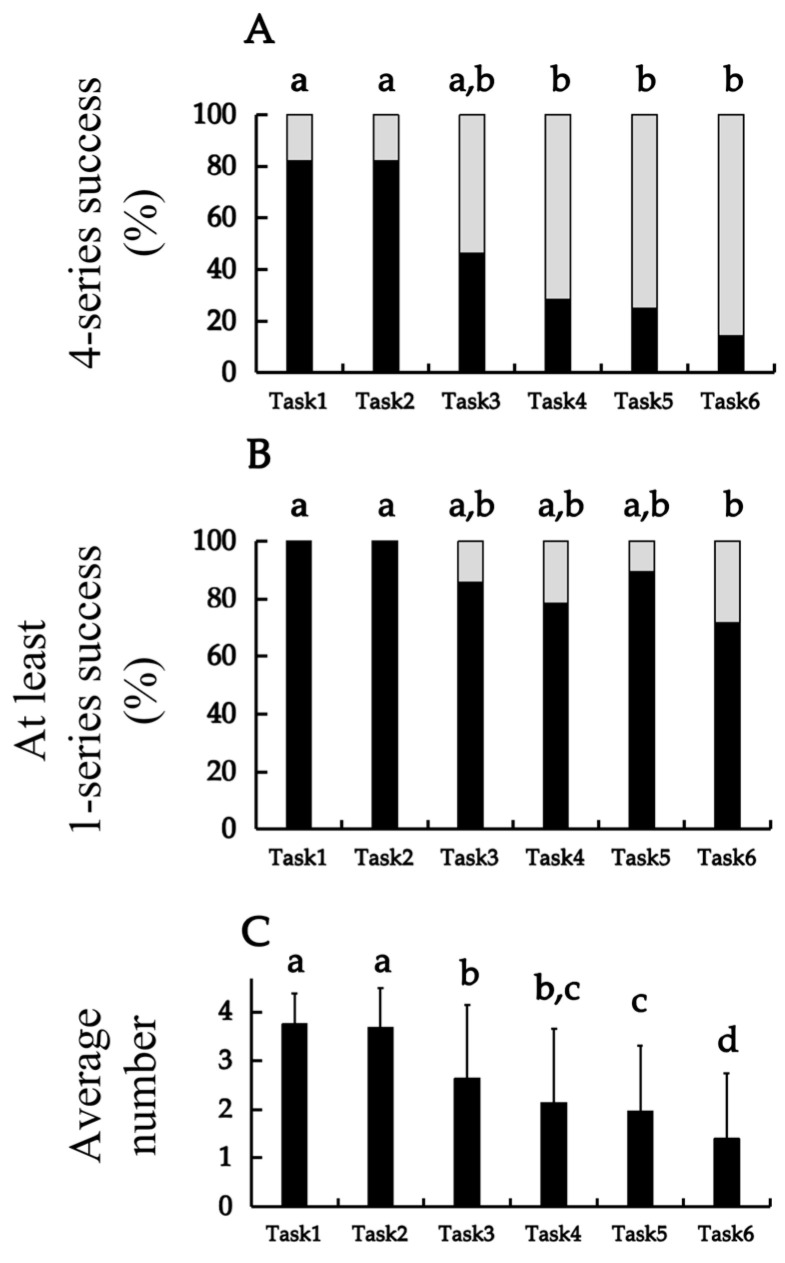
Comparison of the successful completion of the six tasks. (**A**) The percentage of success with four series; (**B**) the percentage of success with at least one series; and (**C**) the average number of successful series. Results of (**A**,**B**) and (**C**) were analyzed using Cochran’s Q test and the paired *t*-test, respectively. Black areas represent successful performance and grey areas represent unsuccessful performance. The significant differences are presented in the form of “a”, “b”, “c”, and “d”. The value indicated by “a” is significantly higher than those indicated by “b”, “c”, and “d”. The value indicated by “b” is significantly higher than those indicated by “c” and “d”. The value represented by “c” is significantly higher than that indicated by “d”. The values indicated by the same letters, such as “a, b” or “b, c”, indicate that no significant difference was detected between the corresponding groups (*p* < 0.05).

**Figure 3 jfmk-10-00473-f003:**
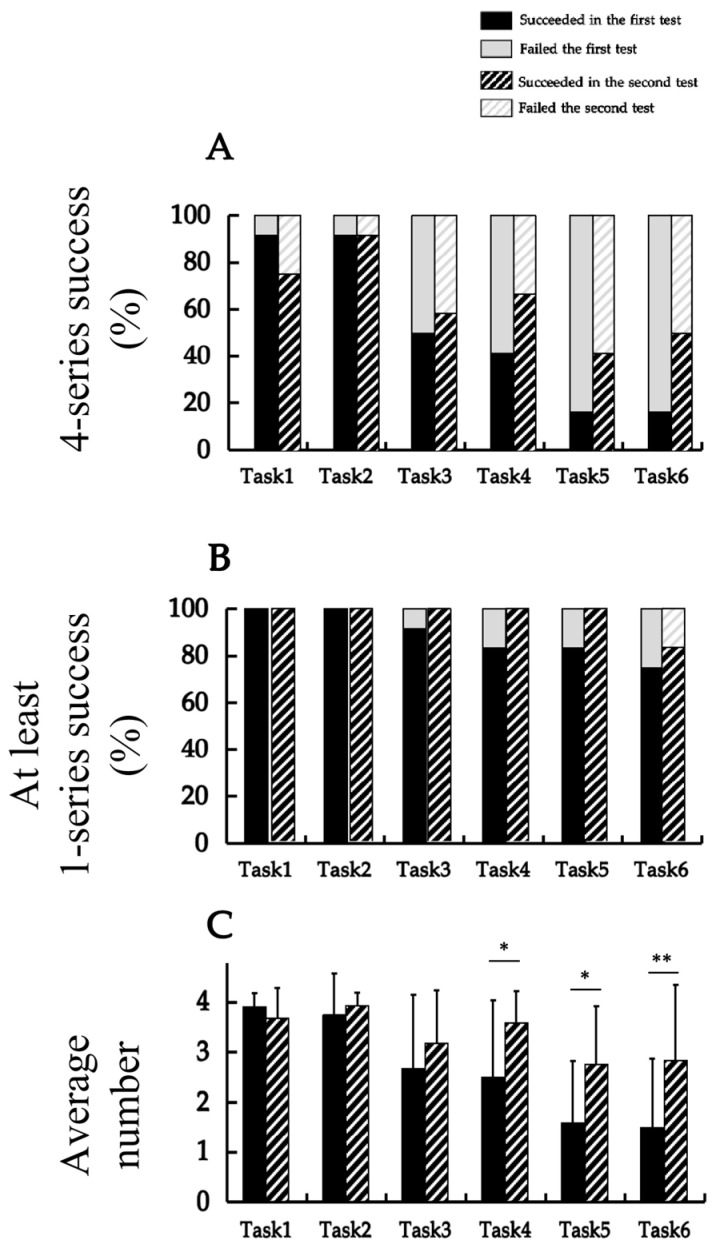
The comparison of the performance between the first and second measurements for the same participants. (**A**) The change in the success rate for all four series, analyzed using McNemar’s test; (**B**) the change in the success rate for at least one series, analyzed using McNemar’s test; (**C**) the change in the average number of successful series, analyzed using the paired *t*-test (* *p* < 0.05; ** *p* < 0.01).

## Data Availability

The data presented in this study are available on request from the corresponding author.

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
