# Peer review of "Validation of a Rhythmic Jumping Task for the Assessment of Upper–Lower Limb Coordination: Evidence from High-Level Athletes"

_jfmk, 2025, doi:10.3390/jfmk10040473_

Round 1
Reviewer 1 Report
Comments and Suggestions for Authors
This is a study investigated usefulness of rhythmic jumping task to assess upper-lower limb coordination for high-level athletes using Japanese high school female volleyball players. Based on these results, the authors concluded that the rhythmic jumping tasks may be a valid, practical and reliable tool to assess upper-lower limb coordination in high-level athletes.
While it is an interesting study that focused on validity and usefulness of simple test to assess upper-lower limb coordination, there are few points which the authors should explain in the manuscript:
- While the previous study was conducted on male baseball players, the present study was conducted on female volleyball players. Although the authors discussed no sex differences in upper-lower limb coordination among adolescents, it is certainly easier to interpret the results from the same sex. To compare sex difference, the authors may recruit female softball players and to compare the results from athletes differ in limb coordination, the authors may recruit male volleyball players. Please explain why the authors recruited female volleyball players that makes interpretation of the results difficult.
- The authors administered six tasks but the manuscript only described task1. Since how each task differs, in terms of length of time and complexity of upper-lower limb coordination, it is suggested the authors provide details of each task in the text as well as in a table. In addition, the authors described that all tasks differ in movement frequency between the upper and lower limbs. This information should be stated in the methodology section.
- It seems the success of each task was justified from an observation of recorded video. Please clarify a number of researchers who involved evaluation of each task and any use of tools such as checklist for an objective evaluation of the successful performance. If the evaluation was conducted by a single researcher or not using any checklist, please consider a possible introduction of subjective bias in the limitation.
Reviewer 2 Report
Comments and Suggestions for Authors
The manuscript titled “Validation of a Rhythmic Jumping Task for the Assessment of Upper–Lower Limb Coordination: Evidence from High-Level Athletes” presents a thematic continuation of a previously published coordination task, now applied to a new sample composed of female volleyball players, distinct from the prior study that evaluated male baseball players. The study uses entirely new data, includes athletes with high coordination levels, and incorporates a one-year retest. The topic is relevant within the field of applied motor control and field-based evaluation, and the work offers incremental evidence regarding the validity and stability of the proposed task. However, some revisions are necessary to strengthen internal validity.
Below are detailed comments:
- Introduction
The introduction is generally well organized but excessively descriptive, with the critical gaps in the literature presented only superficially. It would be valuable for the introduction to address: What do we still not know that truly matters? Which limitation of previous studies prevents further advancement in the field? Why is the present research necessary now? and, in this last point, to address the impacts, whether scientific, economic, or social.
Several paragraphs summarize well-established concepts (motor coordination, rhythm, sensorimotor synchronization), but do not clearly identify the specific limitation that justifies the present study.
The authors should articulate:
- The introduction does not clarify why high-level athletes constitute a necessary methodological step for advancing the investigation, rather than simply representing a distinct and conveniently accessible sample. The authors should explicitly state the scientific relevance of including this group and how it contributes to testing limits, sensitivity, or generalization of the task.
- Which aspects of validity (construct, criterion, discriminant) remain untested.
- Why a one-year retest is important, rather than merely an observational follow-up.
- Methods
2.1 Participants
- The sample is small (n = 28; retest = 12) and lacks justification of statistical power.
- The comparison with the previous study is not a controlled design; the authors must acknowledge the limitations of comparing independent studies.
- The manuscript discusses differences between the current group (female) and the previous group (male), but such differences were neither controlled nor analytically tested.
2.2 Task Structure
- The movement descriptions depend on a previous publication.
- To make the method clearer and more accessible, it is recommended to include a complete operational description of the task within the manuscript itself, even if presented concisely, preventing the reader from needing to fully consult the prior study to understand the procedure.
- The images in Figure 1 have low resolution (see page 4) and do not allow precise interpretation of limb positions.
2.3 Data Processing
The criteria for “success” versus “failure” require greater operational clarity.
- What constitutes a “deviation”?
- How many frames or beats determine failure?
- Was inter-rater reliability assessed?
Reliance on a single evaluator for video analysis introduces risk of classification bias.
Statistical Analysis
- Justify why parametric tests were used for count-based outcomes.
- Report effect sizes consistently.
- Consider and discuss the risk of multiple comparisons.
- Results
- The results are repetitive; the same patterns appear across the three indicators, generating excessive length without proportional informational gain.
- The retest results are based on a small number of participants (n = 12), which limits the statistical power of the analyses and hinders the detection of real differences over time. Thus, although several tests yielded nonsignificant results, interpreting these findings as robust evidence of stability should be approached with caution. It is recommended that the authors explicitly acknowledge this limitation and avoid strong conclusions that cannot be supported by the available sample size.
- Discussion
- The claim that the task demonstrates a “generalizable difficulty structure” is not fully supported, given the absence of multicultural samples, other sports, or adult participants.
- The similarity with the previous study may simply reflect structural characteristics of the task rather than validity across distinct populations.
The discussion does not sufficiently explore:
- Possible learning effects between sessions.
- The possibility that rhythm-based volleyball training may artificially inflate performance relative to baseball players.
- Limitations
The limitations section is appropriate but needs expansion, including:
- Small sample size and substantial attrition (12 of 28 in the retest).
- Absence of randomization or a control group.
- Lack of physiological or biomechanical measures, despite their mention in the introduction.
- Potential bias arising from subjective video-based evaluation.
The manuscript contains specific English-language issues, although overall comprehension is possible. These issues do not render the text unreadable, but they do affect precision, fluency, and academic quality.
Examples include:
- Unnatural sentence constructions:
“Therefore, it is considered that high-level athletes may demonstrate better performance due to their advanced coordination capabilities, and thus the present study aimed to clarify whether…” - Inconsistent use of verb tenses, with unnecessary switching between simple past, simple present, and present perfect:
“was measured and is evaluated…” (mixing past + present)
“the participants are instructed and performed…” (instruction in the present + action in the past) - Subject–verb agreement errors:
“The results indicates…”
